# Striking Cardioprotective Effects of an Adiponectin Receptor Agonist in an Aged Mouse Model of Duchenne Muscular Dystrophy

**DOI:** 10.3390/antiox13121551

**Published:** 2024-12-18

**Authors:** Michel Abou-Samra, Nicolas Dubuisson, Alice Marino, Camille M. Selvais, Versele Romain, Maria A. Davis-López de Carrizosa, Laurence Noel, Christophe Beauloye, Sonia M. Brichard, Sandrine Horman

**Affiliations:** 1Endocrinology, Diabetes and Nutrition Unit, Institute of Experimental and Clinical Research (IREC), Medical Sector, Université Catholique de Louvain (UCLouvain), Avenue Hippocrate 55, 1200 Brussels, Belgium; nicolas.j.dubuisson@uclouvain.be (N.D.); camille.selvais@uclouvain.be (C.M.S.); romain.versele@uclouvain.be (V.R.); mayadavis@us.es (M.A.D.-L.d.C.); laurence.noel@uclouvain.be (L.N.); sonia.brichard@uclouvain.be (S.M.B.); 2Neuromuscular Reference Center, Department of Neurology, Cliniques Universitaires Saint-Luc, Avenue Hippocrate 10, 1200 Brussels, Belgium; 3Pole of Cardiovascular Research, Institute of Experimental and Clinical Research (IREC), Université Catholique de Louvain (UCLouvain), Avenue Hippocrate 55, 1200 Brussels, Belgium; alice.marino@uclouvain.be (A.M.); christophe.beauloye@uclouvain.be (C.B.); sandrine.horman@uclouvain.be (S.H.); 4Departamento de Fisiología, Facultad de Biología, Universidad de Sevilla, 41012 Seville, Spain; 5Department of Cardiovascular Intensive Care, Cliniques Universitaires Saint-Luc, Avenue Hippocrate 10, 1200 Brussels, Belgium

**Keywords:** adiponectin, Duchenne muscular dystrophy, cardiomyopathy, inflammation, fibrosis, mitochondria, AMPK

## Abstract

Adiponectin (ApN) is a hormone with potent effects on various tissues. We previously demonstrated its ability to counteract Duchenne muscular dystrophy (DMD), a severe muscle disorder. However, its therapeutic use is limited. AdipoRon, an orally active ApN mimic, offers a promising alternative. While cardiomyopathy is the primary cause of mortality in DMD, the effects of ApN or AdipoRon on dystrophic hearts have not been investigated. Our recent findings demonstrated the significant protective effects of AdipoRon on dystrophic skeletal muscle. In this study, we investigated whether AdipoRon effects could be extended to dystrophic hearts. As cardiomyopathy develops late in mdx mice (DMD mouse model), 14-month-old mdx mice were orally treated for two months with AdipoRon at a dose of 50 mg/kg/day and then compared with untreated mdx and wild-type (WT) controls. Echocardiography revealed cardiac dysfunction and ventricular hypertrophy in mdx mice, which were fully reversed in AdipoRon-treated mice. AdipoRon also reduced markers of cardiac inflammation, oxidative stress, hypertrophy, and fibrosis while enhancing mitochondrial biogenesis via ApN receptor-1 and CAMKK2/AMPK pathways. Remarkably, treated mice also showed improved skeletal muscle strength and endurance. By offering protection to both cardiac and skeletal muscles, AdipoRon holds potential as a comprehensive therapeutic strategy for better managing DMD.

## 1. Introduction

Duchenne muscular dystrophy (DMD) is a devastating condition characterised by the progressive wasting of skeletal muscles, respiratory insufficiency, and cardiomyopathy [1]. Historically, respiratory failure has been the primary factor contributing to mortality in patients with DMD. However, advancements in the management of respiratory symptoms have increased the life expectancy of DMD patients. Increased longevity has, thus, heightened the clinical importance of heart disease in DMD, with nearly all patients aged 18 and older showing signs of cardiomyopathy [2]. In the last decade, heart failure has even emerged as the primary cause of death in the DMD [3].

Early identification and timely intervention are essential to address cardiomyopathy associated with DMD. Unfortunately, existing treatments for dilated cardiomyopathy remain insufficient due to a limited comprehension of the precise mechanisms contributing to heart failure in DMD. Conventional treatments often include glucocorticoids, angiotensin-converting enzyme inhibitors, and β-adrenoceptor antagonists. Presently, these therapies are only able to slow down the processes of cardiac remodelling and the onset of heart failure without reversing it [4]. Therefore, new medications are highly needed.

For nearly twenty years, our research has focused on the fascinating properties of adiponectin (ApN), with particular emphasis on one of its main target tissues, skeletal muscle [5]. ApN is a hormone released by adipocytes. Its actions include promoting insulin sensitivity, facilitating fat burning, and exerting anti-inflammatory and antioxidative effects. These properties make it a powerful agent for alleviating many metabolic disorders, such as type 2 diabetes, obesity, and cardiovascular disease [5]. The pleiotropic effects of ApN are mediated through its interaction with two primary receptors (AdipoRs), AdipoR1 and AdipoR2 [6]. AdipoR1 is predominantly expressed in skeletal and cardiac muscles, while AdipoR2 exhibits higher expression levels in the liver. This distribution of receptors allows ApN to elicit its beneficial effects across various tissues, contributing to its role in maintaining metabolic balance and combating associated disorders [5].

More recently, we have extended our focus on ApN impacts in the context of DMD. Our research and that of others have revealed a decrease in circulating ApN levels in mdx mice [7,8] and human patients [9]. Notably, ApN supplementation has been shown to counteract disease progression in the best-studied animal model of DMD, namely, the mdx mouse model [7]. These findings support a rationale for targeting low ApN levels in individuals with DMD as a therapeutic strategy. However, the therapeutic application of ApN is significantly restricted [5]. Recently, a number of small molecules and short peptides have been identified, targeting AdipoRs and replicating some of the effects attributed to ApN. This opens up avenues for therapeutic interventions that harness ApN’s beneficial properties through alternative agents with improved applicability and effectiveness [5].

AdipoRon, identified as an AdipoR small-molecule agonist [10], has demonstrated compelling outcomes in addressing dystrophic alterations. Indeed, it exhibited the ability to alleviate muscle inflammation and injury, concurrently promoting muscle regeneration and function [11]. Additionally, AdipoRon has already exhibited protective effects on the heart by attenuating post-ischemic myocardial apoptosis [12] as well as inflammation and cardiac dysfunction after cardiopulmonary bypass [13]. It has also mitigated cardiac inflammation, oxidative stress, and apoptosis by enhancing cardiac lipid metabolism in type 2 diabetic mouse models [13,14]. Yet, there is a significant gap in our understanding of AdipoRon’s impact on Duchenne cardiomyopathy. Exploration of its effects in this specific context is, therefore, of interest.

To this end, AdipoRon was administered orally, at an optimal dose of 50 mg/kg/day [11] for a period of 8 weeks, starting at the age of 14 months, while dilated cardiomyopathy and contractile deviances started to be evident after 12 months of age in mdx mice [15,16]. We first examined whether treated mice showed reduced cardiac dysfunction, reduced inflammation, oxidative stress and fibrosis, and improved mitochondrial function, thus contributing to the attenuation of the dystrophic phenotype. Then, we uncovered the potential mechanisms of action underlying the effects of AdipoRon.

## 2. Materials and Methods

### 2.1. Animals

C57BL/10ScSn-DmdmdxJ mdx mice (murine model of DMD) and C57BL/10ScSnJ mice (used as wild-type (WT) controls) were purchased from Jackson Laboratory (Bar Harbor, ME, USA). Three groups of male mice (n = 6–8/group) were compared in our experiments. At the age of 14 months, each group of mice was given solid drinks, replacing bottled water. The first group comprised WT mice; the second group included untreated mdx mice (mdx), and the third group included treated mdx mice (mdx-AR). For this last group, a dose of 50 mg/kg/day of AdipoRon (Bio-Techne, Minneapolis, MN, USA) was incorporated into a solidified water (Solid Drink, Triple A Trading, Tiel, The Netherlands). Solid drinks were replaced every day for 8 weeks. Animals were fed a standard laboratory chow and housed at a constant temperature (22 °C) with a fixed 12-h light/dark cycle (lights on from 7:00 a.m. to 7:00 p.m.). At the end of the treatment, 16-month-old mice were sacrificed, as described [11], 1 week after echocardiography and in vivo functional tests.

### 2.2. Echocardiographic Analyses

Cardiac dimensions and function were evaluated via transthoracic echocardiography, performed with a Vevo 3100 Imaging System (FUJIFILM VisualSonics, Toronto, ON, Canada), as previously described [17]. Mice were lightly anaesthetised with inhaled isoflurane (1%, in 100% O_2_). Left ventricle (LV) volumes were measured using B-mode, parasternal long-axis view, at end-systole and end-diastole, from which ejection fraction (EF %) was deduced. LV mass was calculated from long-axis measurements. Left ventricle end-diastolic (LVDd) and end-systolic (LVDs) dimensions were measured from the M-mode traces, and fractional shortening (FS %) was calculated as follows: [(LVDd − LVDs)/LVDd × 100]. All measurements and analyses were performed by the same experienced operator who was blinded to the experimental groups.

### 2.3. In Vivo Studies of Global Force or Resistance

Mice were submitted to 3 main tests [11]:

Wire test. Mice were suspended by their limbs from a 1.5 mm-thick, 60 cm-long metallic wire at 45 cm above soft ground. The time (s) until they completely released their grasp and fell was recorded. Mice that reached the 180 s limit were allowed to stop the experiment, while the others were directly retested up to three times, and their maximum suspension time was recorded. For each mouse, three trials were performed at 15 min intervals, and the scores from the three trials were averaged.

Grip test. Mice were gently placed on top of the sensor-connected grid (Panlab-Bioseb, Vitrolles, France), so that their front paws (forelimb test) or fore and hind paws (combined test) could grip the grid. The mice were then pulled steadily backwards until their grip was released along the entire length of the grid. Each test was repeated three times at 15 min intervals. Results are presented as the mean of the 3-force recorded values related to body weight.

Eccentric exercise. Mice were subjected to a downhill running exercise on a treadmill (Panlab-Bioseb) with a 15° incline and at a speed of 10 m/min for 10 min. This training was repeated daily for 3 days. Results represent the distance (in meters) covered by each mouse on day 3, with a maximum distance of 100 m.

### 2.4. Quantification of Muscle Damage Markers in Plasma

Plasma Creatine Kinase (CK) and Lactate Dehydrogenase (LDH) activities were quantified to assess muscle damage, as injured muscles (skeletal and cardiac) released CK and LDH into the bloodstream. CK activity herein did not distinguish between the different CK isoforms. The kits were based on colourimetric methods and were used in accordance with the manufacturer’s instructions (BioAssay Systems—GENTAUR BVBA, Kampenhout, Belgium). CK and LDH activities were expressed in IU/L.

### 2.5. RNA Extraction and Real-Time Quantitative PCR (RT-qPCR)

RNA was isolated from mouse cardiac muscles with TriPure reagent (Sigma-Aldrich, Overijse, Belgium). An amount of 1 µg of total RNA was reverse-transcribed, and 40 ng of total RNA equivalents were amplified using an iCycler iQ real-time PCR detection system (Bio-Rad Laboratories, Nazareth, Belgium), as described [11]. RT-qPCR primers for mouse cyclophilin, tumour necrosis factor-alpha (TNFα), interleukin 1 beta (IL 1β), interleukin 10 (IL-10), peroxiredoxin 3 (PRDX3), adiponectin receptors 1 and 2 (AdipoR1/2), mitochondrial transcription factor A (mtTFA), transforming growth factor beta 1 (TGF-β1), smooth muscle alpha-actin (αSMA), B-type natriuretic peptide (BNP), atrial natriuretic peptide (ANP), peroxisome proliferator-activated receptor-gamma coactivator (PGC-1α), and estrogen-related receptor alpha (ERRα) were used as reported [11,17,18,19,20]. Threshold cycles (Ct) were measured in separate duplicate tubes. The identity and purity of the amplified product were verified by electrophoresis on agarose minigels, and melting curve analysis was performed at the end of amplification. Additionally, each plate included a negative control for each primer set.

### 2.6. Protein Extraction and ELISAs

Mouse heart muscles were homogenised in a lysis buffer supplemented with a 1% protease/phosphatase inhibitor cocktail (Cell Signaling Technology, BIOKE, Leiden, The Netherlands) and 10 mM NaF. Proteins were quantified using the Bradford method and stored at 80 °C. An amount of 25–150 µg of total protein extracts was used for each analysis. ELISA assays quantified the active phosphorylated form of AMP-activated protein kinase alpha (P-AMPK), the active phosphorylated form of the p65 subunit of nuclear factor kappa B (NF-κB) (P-P65), and the active phosphorylated form of SMAD2 (P-SMAD2) (all from Cell Signaling Technology). ELISA assays were also used to quantify the levels of ANP, 4-Hydroxynonenal (HNE), TNFα, IL-1β, IL-10 (all from Abcam, Cambridge, UK), utrophin A (UTRN) (from Antibodies Online, Atlanta, GA, USA), TGF-β, AdipoR1, AdipoR2, the active phosphorylated form of Calcium/Calmodulin Dependent Protein Kinase Kinase 2 (CAMKK2), peroxisome proliferator-activated receptor α (PPARα), PPAR gamma coactivator 1 alpha (PGC-1α), Translocase of Outer Mitochondrial Membrane 20 (TOMM20), and the active phosphorylated form of Ribosome-inactivating protein (P-RIP) (all from MyBiosource—Bio-Connect Diagnostics B.V., Huissen, The Netherlands). The kits were based on colourimetric methods and were used in accordance with the manufacturer’s instructions.

### 2.7. Bright-Field Histochemistry and Morphometry

Heart muscle samples were fixed in 10% formalin for 24 h and then embedded in paraffin. The 5 µm sections were processed as previously described [11], using rabbit polyclonal antibodies directed against TNFα (dilution 1:100, incubation 2 h 30 min), IL-1β (1:200, 2 h 30 min), PRDX3 (1:500, 2 h 30 min), and HNE, (1:100, 2 h 30 min) (PRDX3 is a gift from Bernard Knoops [21], University of Louvain, Brussels, Belgium; the rest are from Abcam). Prior to immunostaining, sections were subjected to hot antigen retrieval using a microwave oven and Tris-citrate buffer (pH 6.5). Antibody binding was detected by applying a secondary antibody for 30 min at room temperature. The secondary antibody used was a biotinylated goat anti-rabbit IgG (H + L) (Labconsult, Brussels, Belgium). Peroxidase activity was revealed using 3,3′-diaminobenzidine (DAB) (Thermo Fisher Scientific, Aalst, Belgium), which produced a brown staining. For each marker, all slides were processed simultaneously for immunohistochemical analysis and DAB revelation, then analysed together. Immunohistochemical controls were performed by omitting the first antibody or the first and second antibodies or by using pre-immunised serum. In addition, Picro-Sirius red (Abcam) staining was used to evaluate muscle fibrosis. Whole muscle sections were scanned using the Leica SCN400 slide scanner (Leica microsystems, Diegem, Belgium), and then the percentage of DAB-deposited or red-stained areas in the muscle fibres was quantified using ImageJ (NIH, Bethesda, MD, USA).

### 2.8. Statistical Analyses

Results are means ± SD for the indicated number of mice. The effects of AdipoRon on the three groups of mice (WT, mdx, and mdx-AR) were analysed using one-way ANOVA, followed by Tukey’s post hoc test, performed with Prism 9 (GraphPad Software, San Diego, CA, USA). Differences were considered statistically significant at *p* < 0.05.

### 2.9. Study Approval

All procedures conducted on mice were approved by the Ethical Committee for Animal Experimentation from the Medical Sector at Université Catholique de Louvain (n° LA1230396).

## 3. Results

### 3.1. Effects of AdipoRon Treatment on Dystrophic Cardiac Dysfunction

Since mdx mice display significant cardiac morphological abnormalities later on in life, and since cardiac dysfunction is only detectable by echocardiogram starting 12 months of age (Refs. [15,16], our own pilot study), AdipoRon was orally administered to 14-month-old mdx mice for a duration of two months. Mdx littermates were, thus, divided into two groups: one treated with AdipoRon at 50 mg/kg (mdx AR) and the other remaining untreated (mdx). Both groups were also compared to untreated wild-type (WT) control mice. Echocardiographic analyses of the three groups of mice showed no significant differences in end-systolic and end-diastolic volumes or fractional shortening (Figure 1A–C). However, the heart rate was increased, and the ejection fraction was decreased in mdx mice compared to WT mice, while they were restored after AdipoRon treatment (Figure 1D,E). Moreover, body weight was slightly decreased only for mdx mice when compared to WT mice (Figure 1F). LV mass tended to increase in mdx mice but was corrected by the treatment (Figure 1G). Finally, LV mass normalised to body weight (BW) was significantly increased in mdx mice compared to WT mice, while it was again normalised after AdipoRon treatment (Figure 1H). These data indicate that AdipoRon treatment limits the development of left ventricular hypertrophy while preserving cardiac function in aged mdx mice.

### 3.2. Effects of AdipoRon Treatment on Dystrophic Cardiac Inflammation and Oxidative Stress

We next tested ex vivo whether oral administration of AdipoRon to aged mdx mice could reverse the dystrophic pathology by counteracting deleterious inflammatory and stress responses in cardiac muscle, as previously seen in the dystrophic skeletal muscle [11]. When compared to WT mice, cardiac muscle fibres of aged untreated mdx mice exhibited elevated expression of TNFα and IL-1β, two genes encoding two major inflammatory cytokines [22], which normalised following AdipoRon treatment (Figure 2A,B). These results were then validated by ELISA, where both TNFα and IL-1β were found to be highly produced in dystrophic cardiac muscles (two-fold increase vs. WT mice) while being decreased under AdipoRon treatment (Figure 2E,F). Moreover, AdipoRon almost doubled the gene expression and protein levels of IL-10, a key anti-inflammatory cytokine [23], when compared to both WT and mdx mice (Figure 2C,G). Lastly, dystrophic cardiac muscle fibres had a ~40% increase in the expression of peroxiredoxin 3 (PRDX3), a marker of oxidative stress (Figure 2D), and a 2.5-fold increase in the protein levels of 4 hydroxy-2-nonenal (HNE), a lipid peroxidation product (Figure 2H) [7,11].

In addition, immunolabelling revealed that these markers of inflammation and oxidative stress were highly produced within cardiac muscle fibres of mdx mice when compared with WT and mdx-AR mice (Figure 3A). Quantification of immunolabelling revealed a strong correlation between the extent of DAB staining for all markers in cardiac muscle fibres and their corresponding mRNA and protein levels (Figure 3B–E). Therefore, AdipoRon demonstrates a robust ability to protect dystrophic cardiac muscle from excessive inflammatory responses and oxidative stress.

### 3.3. Effects of AdipoRon Treatment on Dystrophic Cardiac Fibrosis and Hypertrophy

We also investigated the effects of AdipoRon treatment on markers of fibrosis and hypertrophy in the dystrophic cardiac muscle. First, the mdx muscle showed strong staining for Picrosirius red, a fibrosis marker labelling collagen types I and III. In contrast, the percentage of fibrotic areas was halved in mdx-AR mice (Figure 4A,B). Moreover, and as expected, gene expression for αSMA and TGF-β1, two markers of cardiac fibrosis, was significantly increased by ~2- and 2.5-fold in mdx mice compared to WT mice, respectively, while being reduced by 30–50% under AdipoRon treatment (Figure 4C,D). Similarly, protein levels of active TGF-β and its downstream effector, phosphorylated SMAD2 (P-SMAD2), were elevated approximately 2- and 2.5-fold, respectively, in mdx mice compared to WT mice. However, these levels were reduced by 25–50% in treated mdx-AR mice (Figure 4E,F). Next, gene expression for both BNP and ANP, two markers of hypertrophy, significantly increased by 2.8- and 1.7-fold, respectively, in mdx cardiac muscles compared to WT ones, with ANP protein levels almost doubling (Figure 4G–I). This was then corrected, and all mRNA/protein levels were normalised under AdipoRon treatment (Figure 4G–I). These data indicate a strong protective effect of AdipoRon against cardiac fibrosis and hypertrophy.

### 3.4. Effects of AdipoRon Treatment on Dystrophic Cardiac Signalling Pathways

AdipoRon, like ApN, can bind to AdipoR1 and AdipoR2 and activate similar pathways, such as AMPK and PPARα signalling, respectively [5,6]. We first explored the expression and production of both AdipoRs in the cardiac muscles. We detected a ~30–50% decrease in gene expression and protein levels of AdipoR1 and AdipoR2 in mdx mice compared to WT, which were then corrected under AdipoRon treatment (Figure 5A–D). However, when comparing the CT values, we revealed that AdipoR1 expression was 1000-fold higher than those of AdipoR2 in the cardiac muscle (Figure 5E), making AdipoR1 the primary ApN receptor in the heart. Next, we investigated the main signalling pathways possibly involved in the mechanisms of action of AdipoRon on the dystrophic cardiac muscle. First, no changes were detected in the activity of AMPK between WT and mdx mice. However, in mdx-AR mice, the active phosphorylated form of AMPK was drastically increased by ~2.5-fold (Figure 5F). In addition, AdipoRon treatment significantly increased the levels of CAMKK2 (Figure 5G) and slightly improved those of PPARα (Figure 5H). Consequently, through activation of the CAMKK2/AMPK signalling, AdipoRon helped rescue gene expression and protein levels of PGC-1α, which were found to be blunted in untreated mdx mice (Figure 5I,J). Finally, as previously seen in the dystrophic skeletal muscle [11], activation of the AMPK pathway by AdipoRon led to a significant 40% decrease in the activity of NF-κB (Figure 5K) and a 15% increase in the levels of utrophin A, a dystrophin analogue and a target gene of PGC-1α (Figure 5L), in the dystrophic heart.

### 3.5. Effects of AdipoRon Treatment on Dystrophic Cardiac Mitochondria, Death, and Overall (Skeletal and Cardiac) Muscle Function

Finally, we measured the gene expression of ERRα and mtTFA, two key markers of mitochondrial biogenesis and targets of PGC-1α. We found that AdipoRon treatment significantly increased their expression, as well as the protein levels of TOMM20, a marker of mitochondrial content, which followed the same expression profile (Figure 6A–C). These data suggest that by mainly binding to AdipoR1 and activating CAMKK2-AMPK signalling, AdipoRon could potentially enhance mitochondrial content.

This was ultimately translated by an improved overall force and endurance of treated old mdx mice. Indeed, three in vivo functional tests for skeletal muscles were carried out: the wire test, the grip test, and a treadmill exercise [11,20]. The wire test assesses muscle fatigue and coordination by measuring the duration a mouse can remain suspended on a horizontal wire. Mdx mice exhibited significantly reduced hanging times compared to WT mice (31 s vs. 97 s, respectively), whereas mdx-AR mice displayed intermediate resistance, with an average hanging time of 55 s (Figure 6D). The grip test evaluates the strength of limb muscles. Forelimb force in mdx mice was 40% lower than in WT mice but was partially restored by AdipoRon treatment, showing an improvement of over 20% (Figure 6E). Similar results were observed for the combined fore- and hind-limb force (Figure 6F). Finally, resistance to fatigue was further assessed using an eccentric treadmill test. By the third and final day of exercise, both WT and mdx-AR mice achieved the maximum distance of 100 m, whereas the running distance of mdx mice markedly fell to ~75 m (Figure 6G). These data indicate an improvement in dystrophic skeletal muscle strength, coordination, and endurance under AdipoRon treatment in aged mdx mice, beneficial effects to which enhanced cardiac function may help contribute.

Moreover, AdipoRon treatment protected dystrophic muscle from injury, as indicated by reduced plasma activities of creatine kinase (CK) and lactate dehydrogenase (LDH). CK and LDH are sarcoplasmic enzymes that are released in circulation after sarcolemmal breach or tear following injury [7,11,24]. Plasma CK and LDH activities, measured here, were ~four-fold higher in mdx mice than in WT mice, while they declined by more than 40% in treated mdx-AR mice. (Figure 6H,I). Moreover, the active phosphorylated form of RIP protein (P-RIP) measured in dystrophic cardiac muscle was almost 3.5-fold higher than in WT ones but was reduced by 35% after treatment (Figure 6J). P-RIP is a main factor of necroptosis, a programmed form of necrosis or inflammatory cell death, which contributes to myofibre degeneration in DMD muscles [20,25]. Therefore, late AdipoRon treatment appears to significantly mitigate sarcolemmal injury and cell death in aged dystrophic skeletal and cardiac muscles.

## 4. Discussion

The findings of this study demonstrated that an eight-week oral administration of AdipoRon significantly protected the dystrophic cardiac muscle in aged mdx mice. This intervention reduced exaggerated inflammatory responses, fibrosis, and hypertrophy while enhancing mitochondrial biogenesis and rescuing cardiac function. Skeletal muscle function was improved as well, even after short-term treatment in old mice, which is also an original finding under AdipoRon. These favourable outcomes collectively contributed to the amelioration of the dystrophic phenotype.

Evidence suggests that Duchenne dilated cardiomyopathy (DCM) begins with sarcolemmal damage and impaired repair mechanisms, which are subsequently exacerbated by inflammatory cells and cytokines, ultimately leading to necrosis and fibrosis [26,27]. In mdx mice, DCM is marked by accumulation of connective tissue and alterations in contractility, the latter developing as early as 10–12 months of age [15,28,29]. Our pivotal study (not published) did not show any cardiac echography alteration before 12 months of age. Based on these data, we decided to test 14-month-old mice for the current study. At this old age, the contractile dysfunction of the left ventricular myocardium is present, thereby decreasing the left ventricular ejection fraction [15,30]. These abnormalities were confirmed in our mdx untreated mice but were normalised after AdipoRon treatment. Morphologically, mdx DCM is characterised by hypertrophy of the left ventricle (LV), including an increase in the size of individual cardiomyocytes. This can be a compensatory response to the increased workload on the heart [31,32]. In our study, LV mass normalised to body mass was higher in mdx than in control mice. We did not, however, find a difference in the LV shortening fraction among the three groups of mice, as has been observed in another study [33]. Additionally, ex vivo testing revealed that markers of hypertrophy were only present in the dystrophic cardiac muscles of untreated animals. Indeed, levels of BNP and ANP, two key genes whose expression is significantly upregulated in pathological cardiac hypertrophy [34], were markedly increased in the hearts of mdx mice when compared to WT. All these abnormalities were again normalised after AdipoRon treatment, indicating a potent protective role against cardiac hypertrophy and dysfunction.

Furthermore, mdx mice treated with AdipoRon showed an elevation in anti-inflammatory cytokines, combined with a reduction in the expression of pro-inflammatory cytokines and oxidative stress markers. The extension of muscle injury and myonecrosis was also diminished. More specifically, levels of CK and LDH, both indicative of muscle membrane breach [11], were reduced, along with P-RIP protein, a key necroptosis factor [20]. This suggests that suppression of RIP activity might, at least partially, contribute to the anti-myonecrotic effect of AdipoRon. Overall, our findings are consistent with other studies showing that AdipoRon administration significantly reduces inflammation, oxidative stress, and tissue damage following heart injury [14].

Fibrosis emerges as one of the initial clinical manifestations in dystrophic cardiac pathology, manifesting in patients before the age of 10 [35] and as early as 2 months of age in mdx mice [36]. It is well-established that cardiac injury and inflammation in DMD trigger the recruitment of fibro-adipogenic progenitors (FAPs), which then differentiate into fibroblasts under the influence of TGF-β, leading to increased deposition of connective tissues [37,38]. TGF-β serves as a pivotal mediator of the fibrotic response, acting through its effector, the phosphorylated and active form of Smad2 [20]. In this study, we highlight the robust anti-fibrotic effects of AdipoRon, as evidenced by a significant reduction in Picrosirius Red staining. Additionally, the quantification of active TGF-β and P-Smad2 revealed substantial decreases following treatment. These findings align with the beneficial effects of AdipoRon observed both on myocardial fibrosis in heart failure with preserved ejection fraction (HFpEF) [39] and intracardial fibrosis in diabetic mice [14].

Both our research and that of others have demonstrated that mdx mice exhibit low circulating levels of ApN [7,8]. In this study, we highlight a significant downregulation of both ApN receptors in the dystrophic cardiac muscles. These findings are consistent with those observed in life-threatening conditions associated with muscle wasting, such as chronic heart failure, where skeletal muscle ApN resistance has been attributed to the downregulation of AdipoR1 and inactivation of AMPK signalling [40]. Herein, AdipoRon treatment restored cardiac AdipoR levels and activated downstream signalling pathways. Indeed, in exploring the mechanisms of action of AdipoRon in the heart, we found that the AMPK signalling pathway was activated in treated mice, as evidenced by increased levels of CAMKK2 and phosphorylated AMPK, combined with subsequent increased PGC-1α levels, as previously observed in dystrophic skeletal muscle [7,11]. Treatment even augmented levels of the AdipoR2-PPARα pathway. This is consistent with observations reported in the hearts of diabetic mice, where AdipoRon was able to increase CAMKK2, AMPK, and PPARα pathways and downstream signalling, thereby decreasing oxidative stress and apoptosis and exerting cardioprotective effects [41].

The AMPK pathway plays a critical role in mediating anti-inflammatory and anti-fibrotic effects while enhancing mitochondrial biogenesis. Firstly, its anti-inflammatory effects are orchestrated primarily through potent suppression of NF-κB signalling [5,20,42]. In our study, there was a notable reduction of over 40% in NF-κB activation. This repression of NF-κB is expected to contribute to the restoration of cardiac function in mdx mice, as demonstrated by another group [16]. Secondly, its anti-fibrotic properties are largely mediated through the inhibition of TGF-β/Smad signalling [17,20,43,44]. For instance, AMPKα1 deletion was found to exacerbate fibrosis in the context of myocardial injury [17], while AMPK activation had an effective anti-fibrotic effect in DMD [20]. Furthermore, the activation of the AMPK pathway has also demonstrated the ability to restrain dysfunctional vascular growth, a significant contributor to cardiovascular disease, by inhibiting TGF-β [45]. This mechanism may contribute to normalising ejection fraction and LV mass (improved cardiac function and diminished cardiac hypertrophy) in our treated mice. It is worth noting that macrophages also contribute to fibrosis in DMD, while AMPK activation may reduce their secretion of TGF-β, thereby attenuating their pro-fibrotic effects [46]. Thirdly, the enhancement of mitochondrial biogenesis has been shown to be mediated by AMPK-PGC-1α [18,19,47]. PGC-1α was even found to stimulate the generation of mitochondria that are geared towards ATP production and exhibit high-level coupled respiration in cardiac myocytes [48]. In the present study, the administration of AdipoRon significantly increased the expression of ERRα and mtTFA, both recognised markers of mitochondrial biogenesis, as well as protein levels of TOMM20, a marker indicative of mitochondrial content. Our findings are consistent with studies showing that AMPK activation or AdipoRon treatment mitigates L-Thyroxine- or isoproterenol-induced cardiac hypertrophy by enhancing myocardial mitochondrial energy metabolism pathways [49,50]. In addition, drug activation of the AMPK-PGC-1α axis restores the mitochondrial network and reduces ROS generation in human foetal fibroblasts with chromosome 21 trisomy [51]. Comparable outcomes were also observed with the overactivation of PGC-1α through histone deacetylase inhibitors, demonstrating mitigation of the dystrophic phenotype by addressing the deficit in mitochondrial biogenesis [52]. Lastly, activation of CAMKK2-AMPK-PGC-1α pathways also resulted in an elevation in utrophin, an analogue of dystrophin, with a noteworthy 15% increase in protein levels. The upregulation of utrophin holds the potential to reinstate sarcolemmal integrity, contributing to both morphological and functional enhancements in mdx mice [53,54]. Our findings demonstrate that AdipoRon acts as a potent AdipoR agonist, inhibiting NF-κB and TGF-β activities while promoting mitochondrial biogenesis and upregulating utrophin. These effects may reduce muscle inflammation, oxidative stress, and fibrosis, protect the heart from injury, and improve overall function, thereby strikingly alleviating the dystrophic phenotype.

Despite significant advancements and extensive research during the past three decades, an effective treatment for DMD is not available yet [55]. Various pharmacological approaches, including glucocorticoids, angiotensin-converting enzyme (ACE) inhibitors, angiotensin receptor blockers (ARBs), mineralocorticoid receptor antagonists, and even simvastatin, have exhibited the potential to curtail fibrosis accumulation in dystrophic mouse and patients’ hearts. However, a prominent anti-fibrotic treatment for DMD remains elusive [56]. Additionally, some of these treatments demonstrate relatively poor efficacy in addressing the specific challenges presented by the dystrophic heart. Hence, there is a crucial need for the development of treatments that strategically address both skeletal and cardiac muscle, and AdipoRon, an ApN receptor agonist, unquestionably emerges as a noteworthy candidate for consideration. Such interventions are crucial not only for enhancing the quality of life but also for extending the longevity of individuals with DMD.

This study unequivocally demonstrated the protective properties of AdipoRon on the dystrophic cardiac muscle. Combined with its beneficial effects on the dystrophic skeletal muscle [11], AdipoRon emerges as a potentially appealing therapeutic option. Furthermore, not only the safety profile of AdipoRon has been corroborated in various murine models and toxicology studies, but it has also been demonstrated that AdipoRon even exhibits the added advantage of safeguarding the liver and the kidneys and enhancing metabolic functions in mice [5,10,18,41]. It could even prolong the lifespan of obese or aged treated mice [18,19] and exert various pleiotropic on a variety of tissues and organs [5].

Emerging therapeutic strategies, including gene therapy approaches such as micro-dystrophin cDNA transfer, are widely regarded as the most promising avenues for the treatment of DMD. Preclinical data indicate that micro-dystrophin gene transfer using adeno-associated virus (AAV) can significantly reduce muscle pathology, improve skeletal and cardiac function, and enhance survival in various animal models of DMD [57,58]. However, this therapy has shown limited efficacy thus far, with suboptimal product expression in muscle, highlighting the need for combinatorial treatments [59,60]. An ApN mimic like AdipoRon could offer complementary benefits to enhance gene therapy. By alleviating the dystrophic phenotype in both skeletal and cardiac muscles, AdipoRon could support the long-term restoration of micro-dystrophin, maximising the therapeutic potential of AAV-based gene therapy.

## 5. Conclusions

In conclusion, by mitigating cardiac inflammation, fibrosis, and hypertrophy while enhancing mitochondrial biogenesis and overall performance, AdipoRon addresses key pathological features of the dystrophic phenotype. These findings underscore its potential as a dual-targeted treatment for both skeletal and cardiac muscle complications in DMD, offering a foundation for future clinical translation aimed at improving patient outcomes and quality of life.

## Figures and Tables

**Figure 1 antioxidants-13-01551-f001:**
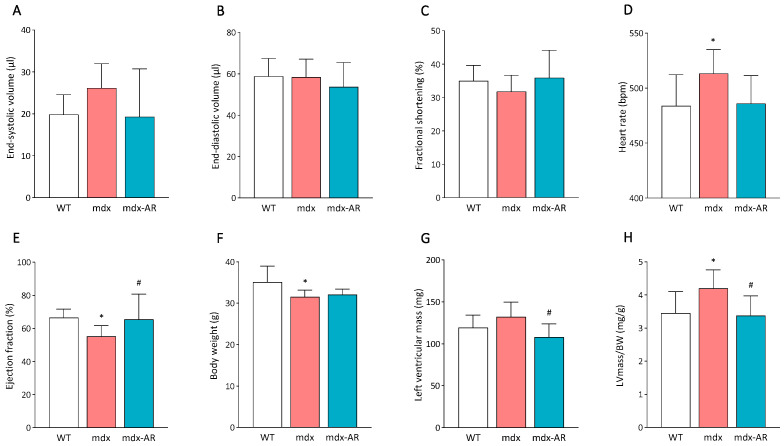
Effects of AdipoRon treatment on cardiac dysfunction in adult mdx mice. Cardiac dimensions and function were analysed in vivo by transthoracic echocardiography in mice from the three groups. (**A**) End-systolic volume (µL). (**B**) End-diastolic volume (µL). (**C**) Fractional shortening (%). (**D**) Heart rate (bpm). (**E**) Ejection fraction (%). (**F**) Body weight (g). (**G**) Left ventricular mass (mg). (**H**) Left ventricular mass over body weight (LVmass/BW) (mg/g). Data are means ± SD; n = 7–8 mice per group for all tests. Statistical analysis was performed using one-way ANOVA followed by Tukey’s test. * *p* < 0.05 vs. WT mice. # *p* < 0.05 vs. mdx mice.

**Figure 2 antioxidants-13-01551-f002:**
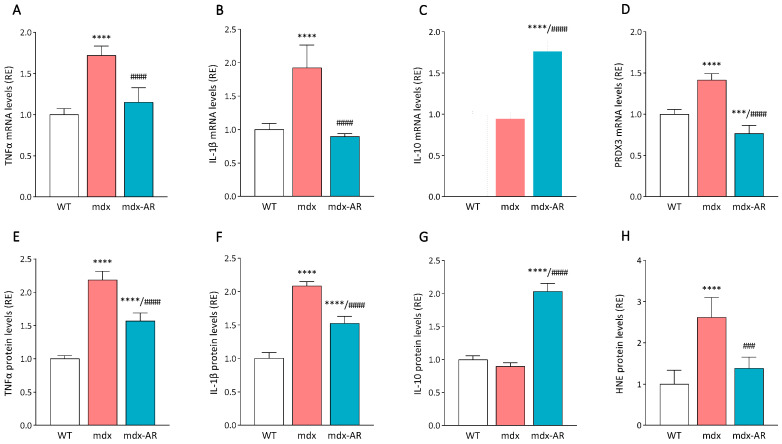
Effects of AdipoRon treatment on cardiac muscle inflammation and stress. mRNA levels of (**A**) TNFα and (**B**) IL-1β, two major inflammatory genes. (**C**) mRNA levels of IL-10, an anti-inflammatory gene. (**D**) mRNA levels of PRDX3, an oxidative stress marker. mRNA levels were normalised to cyclophilin, and the following ratios are presented as relative expressions to WT values. ELISA assays were used to quantify the levels of (**E**) TNFα and (**F**) IL-1β, two major inflammatory cytokines, (**G**) IL-10, a strong anti-inflammatory cytokine, and (**H**) HNE, a lipid peroxidation product. For ELISAs, absorbance data are presented as relative expressions to WT values. Data are means ± SD; n = 6 mice per group for all experiments. Statistical analysis was performed using one-way ANOVA followed by Tukey’s test. *** *p* < 0.001, **** *p* < 0.0001 vs. WT mice. ### *p* < 0.001, #### *p* < 0.0001 vs. mdx mice.

**Figure 3 antioxidants-13-01551-f003:**
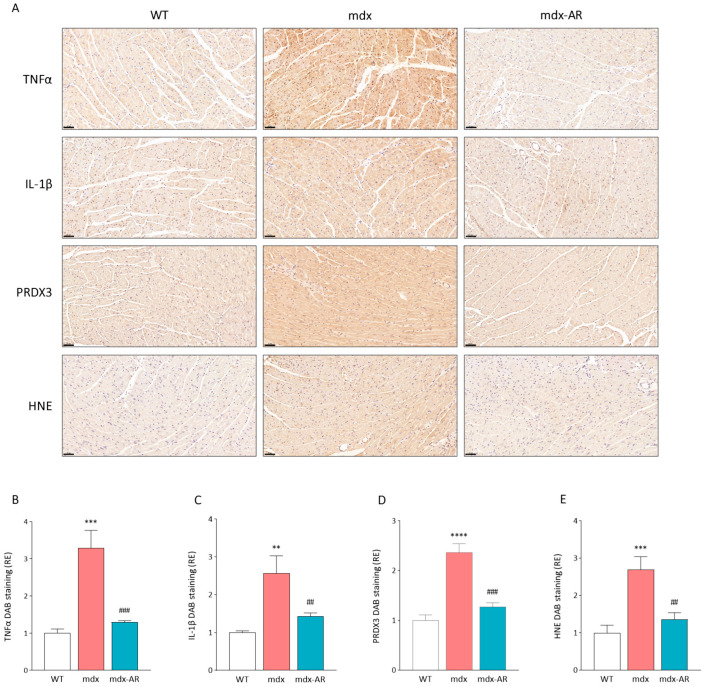
Effects of AdipoRon treatment on cardiac muscle inflammation and stress. (**A**) Immunohistochemistry was performed on cardiac muscle sections with specific antibodies directed against two pro-inflammatory cytokines (TNFα and IL-1β) and two oxidative stress markers (PRDX3 and HNE). Scale bar = 50 μm. Quantification of (**B**) TNFα, (**C**) IL-1β, (**D**) PRDX3, and (**E**) HNE. For each immunolabelling of (**A**), the percentage of DAB deposit areas was calculated in cardiac muscle sections. The subsequent ratios are presented as relative expressions to WT values. Data are means ± SD; n = 6 mice per group for all experiments. Statistical analysis was performed using one-way ANOVA followed by Tukey’s test. ** *p* < 0.01, *** *p* < 0.001, **** *p* < 0.0001 vs. WT mice. ## *p* < 0.001, ### *p* < 0.001 vs. mdx mice.

**Figure 4 antioxidants-13-01551-f004:**
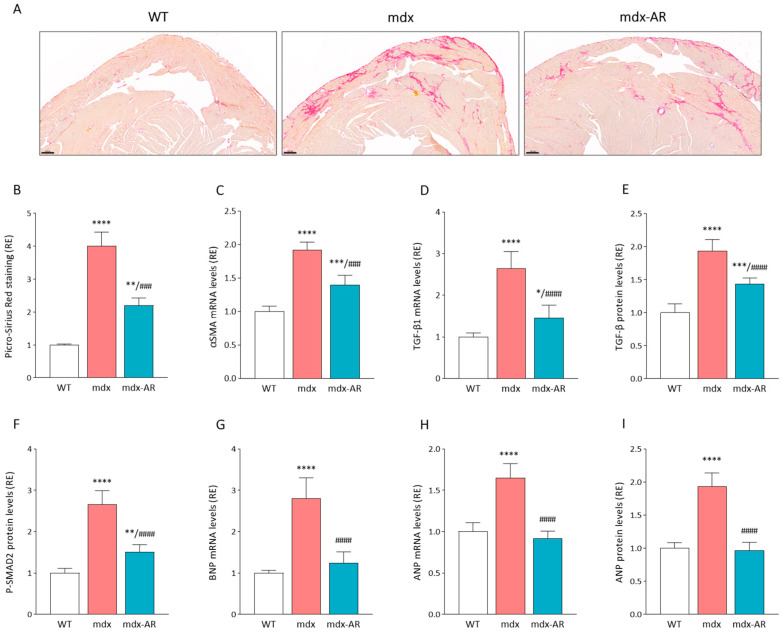
Effects of AdipoRon treatment on cardiac muscle fibrosis and hypertrophy. (**A**) Picro-Sirius Red staining. Scale bar = 200 µm. (**B**) Quantification of Picro-Sirius Red staining. mRNA levels of (**C**) αSMA and (**D**) TGF-β1, two markers of fibrosis. ELISA assays were used to quantify (**E**) TGF-β and (**F**) the active phosphorylated form of SMAD2 (P¬SMAD2), a transcription factor mainly involved in TGF-β signalling. mRNA levels of (**G**) BNP and (**H**) ANP, two markers of hypertrophy. (**I**) ELISA assay was also used to quantify the levels of ANP. The percentage of stained areas was calculated in cardiac muscle sections, and the subsequent ratios are presented as relative expressions to WT values. mRNA levels were normalised to cyclophilin, and the subsequent ratios were presented as relative expressions to WT values. Absorbance data are presented as relative expressions to WT values. Data are means ± SD; n = 6 mice per group for all experiments. Statistical analysis was performed using one-way ANOVA followed by Tukey’s test. * *p* < 0.05, ** *p* < 0.01, *** *p* < 0.001, **** *p* < 0.0001 vs. WT mice. ### *p* < 0.001, #### *p* < 0.0001 vs. mdx mice.

**Figure 5 antioxidants-13-01551-f005:**
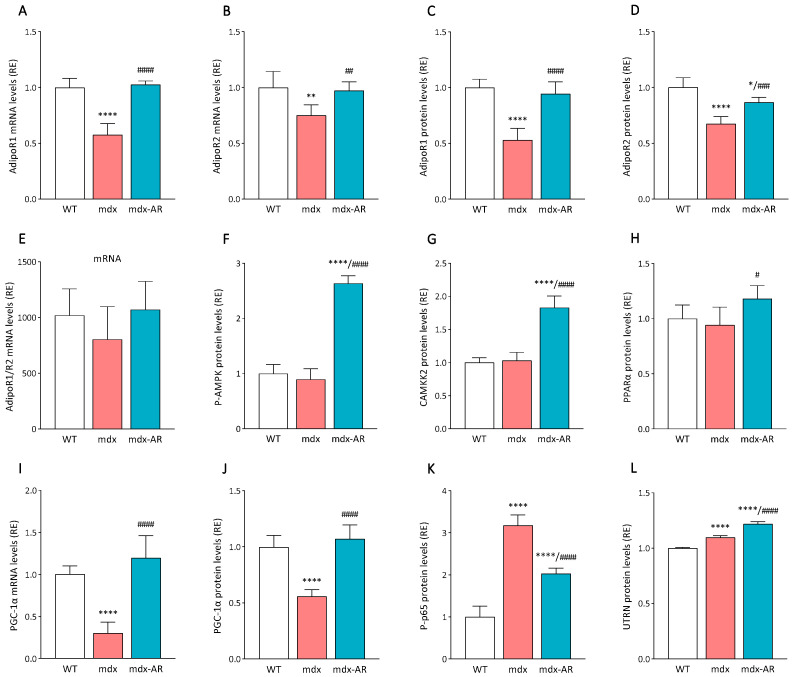
Effects of AdipoRon treatment on ApN receptors and signalling in the dystrophic cardiac muscle. mRNA levels of (**A**) AdipoR1 and (**B**) AdipoR2, adiponectin main receptors. ELISA assays were used to quantify (**C**) AdipoR1 and (**D**) AdipoR2. (**E**) The ratio of AdpoR1 over AdipoR2 mRNA levels was calculated within the cardiac muscle. ELISA assays were used to quantify (**F**) the active phosphorylated form of AMPKα (P-AMPK), (**G**) calcium/calmodulin-dependent protein kinase 2 (CAMKK2), and (**H**) peroxisome proliferator-activated receptor alpha (PPARα), ApN/AdipoRon, main signalling pathways in muscle. (**I**) mRNA levels of PGC-1α. ELISA assays were used to quantify (**J**) PGC-1α, (**K**) the active phosphorylated form of the p65 subunit of NF-κB (P-p65), a transcription factor mainly involved in inflammation, and (**L**) utrophin A (UTRN), a dystrophin analogue. mRNA levels were normalised to cyclophilin, and the subsequent ratios are presented as relative expression to WT values. Absorbance data are presented as relative expressions to WT values. Data are means ± SD; n = 6 mice per group for all experiments. Statistical analysis was performed using one-way ANOVA followed by Tukey’s test. * *p* < 0.05, ** *p* < 0.01, **** *p* < 0.0001 vs. WT mice. # *p* < 0.05, ## *p* < 0.01, ### *p* < 0.001, #### *p* < 0.0001 vs. mdx mice.

**Figure 6 antioxidants-13-01551-f006:**
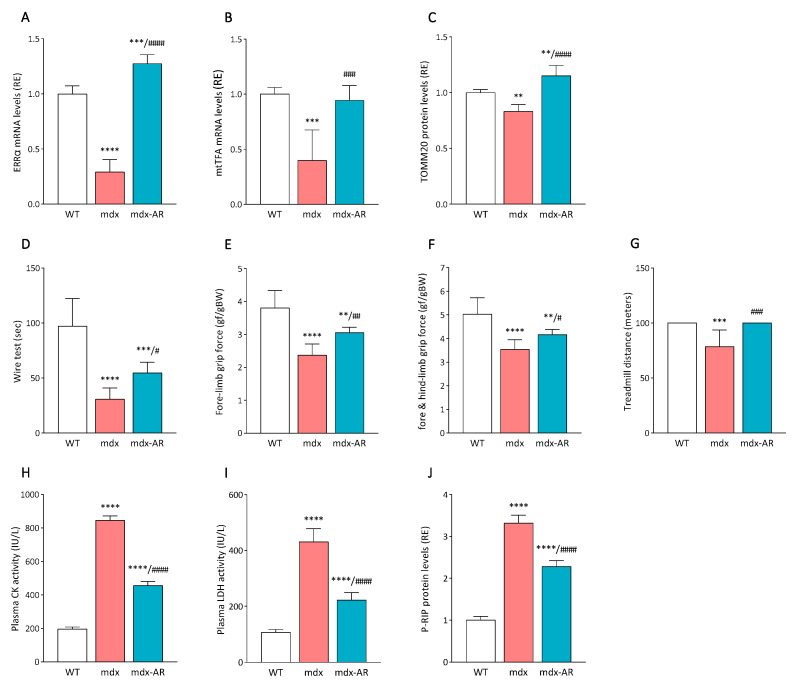
Effects of AdipoRon treatment on cardiac muscle oxidative capacity, injury, and overall muscle function. mRNA levels of (**A**) ERRα and (**B**) mtTFA, two markers of mitochondrial biogenesis. ELISA assay was used to quantify (**C**) TOMM20, a marker of mitochondrial content. (**D**) Wire test where mice hanging time was recorded (s). (**E**) Fore-limb grip test and (**F**) fore- and hind-limb grip test, measuring muscle strength expressed in Gram-force relative to body weight (gf/gBW). (**G**) Treadmill running exercise, where the total distance covered on the third day was measured (m). (**H**) CK and (**I**) LDH plasma activities assessing muscle injury and expressed as IU/L. (**J**) ELISA assay was used to quantify the active phosphorylated form of RIP (P-RIP), an important regulator of cellular stress that triggers a regulated pathway for necrotic cell death called necroptosis. mRNA levels were normalised to cyclophilin, and the subsequent ratios are presented as relative expressions to WT values. Absorbance data are presented as relative expressions to WT values. Data are means ± SD; n = 6 mice per group for all ex vivo experiments. Data are means ± SD; n = 7–8 mice per group for all in vivo functional tests. Statistical analysis was performed using one-way ANOVA followed by Tukey’s test. ** *p* < 0.01, *** *p* < 0.001, **** *p* < 0.0001 vs. WT mice. # *p* < 0.05, ## *p* < 0.01, ### *p* < 0.001, #### *p* < 0.0001 vs. mdx mice.

## Data Availability

All of the data are contained within this article. The raw data supporting the conclusions of this article will be made available by the authors without undue reservation.

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
