# Peer review of "Striking Cardioprotective Effects of an Adiponectin Receptor Agonist in an Aged Mouse Model of Duchenne Muscular Dystrophy"

_antioxidants, 2024, doi:10.3390/antiox13121551_

Round 1
Reviewer 1 Report
In this manuscript Abou-Samra and other Authors describe AdipoRon, an ApN receptor agonist, effects on cardiac muscle of mice affected by Duchenne Muscular Distrophy disease. The same Authors have already reported beneficial and protective results of Adiponectin (ApN) on skeletal muscle, a primary target disease tissue of the same mice strain.
DMD has several phenotypic manifestations, including respiratory insufficiency and heart failure, emerging during the last decade as the primary cause of death in patients. Existing treatments for dilated cardiomyopathy associated with DMD are insufficient due to a limited comprehension of the precise mechanisms.
ApN is a hormone released by adipocytes, exerting anti-inflammatory and antioxidative effects, mediated through its interaction with two primary receptors, AdipoR1 and AdipoR2.
Both in mdx mice and in DMD patients, low circulating levels of ApN together with a significant downregulation of both ApN receptors in the dystrophic cardiac muscles have been described.
Since the use of ApN as a therapeutic agent is severely limited, Authors decided to use Adiporon, the First Orally Active Adiponectin Receptor Activator.
AdipoRon demonstrated to be able to reduce cardiac dysfunction, inflammation, oxidative stress, and fibrosis as well as to enhance mitochondrial function. Moreover, skeletal muscle function was improved. In a second phase of the study Authors also uncovered the potential mechanisms of action underlying the effects of AdipoRon.
The work is interesting and well-designed. I am curious to know if, by sacrificing the mice, they have checked the liver and kidneys to avoid adverse events in "filtering" tissues.
Some considerations:
-I don’t understand why they compare treated and untreated mdx mice with untreated wild type, avoiding to also treat them. I think it is important to observe the therapeutic effect on wt mice.
-In the meantime, an interesting paper was published on Human Gene Therapy by Stephen Baine.
The work was conducted on mdx mice demonstrating long-term cardiac efficacy and improved survival using delandistrogene moxeparvovec treatment. The treatment consists in a gene transfer therapy that uses an adenoassociated viral vector to deliver a micro-dystrophin transgene to skeletaland cardiac muscle. I think that this paper has to be cited and discussed by Authors in their manuscript
-why di they refer and discuss metmorfin in the discussion, without having cited the drug before in the manuscript?
Reviewer 2 Report
The authors have demonstrated convincingly that Adiporon alleviates the fibrosis and other abnormalities in the hearts of DMD mice. The paper is a follow up to a demonstration of the effects of Adiporon on DMD skeletal muscle. A very important aspect of this paper is the use of aged (14 month) mice, where the cardiac abnormalities are overt in DMD mice. The 8 week treatment reverses most of the abnormalities in fibrosis and signalling pathways; this is remarkable since at 14 months one might have supposed the fibrosis to be irreversible. Perhaps the authors could comment on this. It also leads to a question of whether Adiporon treatment of younger mice might prevent the appearance of cardiac abnormalities in DMD mice.
Comment on reversibility of fibrosis
Comment on possibilities of prophylactic treatment with adiporon
